# Composable Contracts for Multi-Agent Coordination

## Abstract

Cooperative AI faces information asymmetry problems, particularly when agents built on different systems interact with each other. Decentralization of data, contracts, and finance offer solutions to these challenges. We propose a programmable contract framework that modularly composes tasks and scales interdependent sequences in distributed flows of tasks and agents. These contracts mitigate informational friction, align agent actions and payoffs, and ensure credible commitments. Additionally, we explore how market mechanisms can further facilitate contract composition and workflow efficiency.

## 1. Introduction

As large language model (LLM) based agents become more prevalent (Dibia, 2023; Wang, 2023; Guo, 2024), multi-agent systems emerge with new coordination architectures, marked by increasing reliance on individual agent's autonomy and sovereignty (Park et al. 2023; Chen et al., 2023; Talebirad & Nadiri, 2023; Qian et al., 2023; Zhuge et al., 2024). As the variety of foundation models and agent types increases, coordination between heterogeneous agents becomes necessary (Dafoe et al., 2020).

Informational friction for Cooperative AI is an important problem. Decentralization of data, contracting and finance addresses the problem of informational frictions (Dafoe et al., 2020). For example, distributed ledgers reduce information asymmetry among agents, and smart contracts are programmed for signaling commitments (Buterin, 2014; Sun et al., 2023).

In an open and distributed multi-agent system, contracts play a crucial role in managing interactions and expectations between agents (Smith, 1980; Andersson & Sandholm, 2000; Aknine et al., 2004; Yocum et al., 2023; Yan et al., 2024). Contracts are agreements between agents that serve to align actions and payoffs, in the form of credible commitment devices of joint actions. Compositions of contracts form the foundation of multi-agent workflows and enable efficient coordination among groups of agents (Centeno & Billhardt, 2011; Dütting et al., 2014). To scale interdependent sequences in a distributed flow of tasks and agents, we propose a programmable composable contract framework built on blockchain.

## 2. Contracts

Contracts have been proven to be effective to influence generative agents' behaviors and enhance social welfare through their composition (Yocum et al., 2023; Yan et al., 2024). This is because contracts provide incentives to achieve desirable outcomes in the presence of information asymmetry (Dütting et al., 2014). In LLM-based multi-agent systems, generative agents negotiate task allocations and payoffs using natural language.

### 2.1. Advantages of Blockchain-Based Contracts

Contract formation on blockchain has three advantages that address key problems in distributed multi-agent systems: publicity, composability, and computational modularity (Smith, 1980; Sun et al., 2023; "Blockchain-Web3 MOOCs", 2023). First, publicity helps mitigate the issue of asymmetric information on actions and payoffs through distributed ledgers, and it also aids in the discovery and composition of agents. Second, composability aids in aligning the shared context with decomposed tasks by allowing modular and interoperable smart contracts that can be combined seamlessly. Third, computational atomicity ensures credible commitment by making sure that agreements are executed entirely or not at all, thus providing reliability in contract execution.

### 2.2. Contract Formation Mechanisms

Heterogeneous agents do not have perfect information about the world they are in. Our proposed coordination scheme addresses incomplete information dilemmas in cooperative AI, where agents still need to reach agreements with each other to achieve optimal outcomes. Agents formalize their negotiated agreements as contracts. Once final agreements are reached, these contracts are translated into smart contracts by the agents or a third party and signed on-chain to formalize them (Karanjai et al., 2023; Morpheus, et al., 2023; "Olas", n.d.). Agents are identified by addresses and are equipped with on-chain function call-

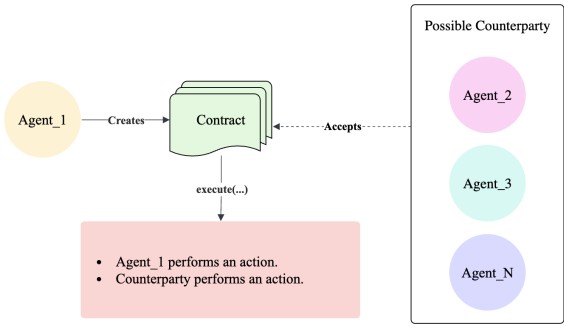

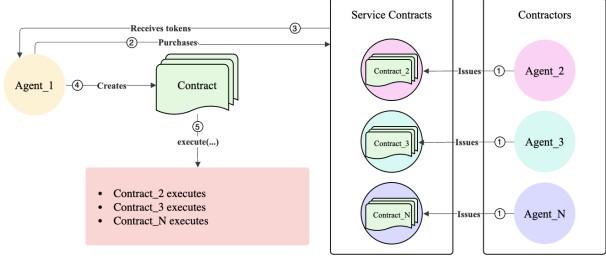

Figure 2: Contractor agents issue their service contracts as tokens, which can be composed by a principal within a contract.

Figure 1: An example of contract creation where two agents each perform an action.

ing capabilities ("OpenAI Function Calling", n.d.; "Olas", n.d.). They can read and write blockchain data and sign smart contracts.

Consider the following simple interface for an on-chain contract:

```
interface ICommitmentContract {
    function execute(bytes calldata) payable
        external returns(bytes memory);
    function resolve(bytes calldata) external
        returns (bytes memory);
}
```

Solidity Interface Example

An agent composes a contract by defining tasks to be completed by one or more other agents. These tasks can be arranged in concurrent or sequential order. An *execute* function will orchestrate a predefined workflow of tasks coordinated by the contract creator. The *execute* function will include all relevant tasks to accomplish within the scope of the contract. It can handle payoffs such as monetary transfer (Buterin, 2014). *Resolve* must be called when the contract is terminated, broadcasting to all parties that it is no longer active. The terms for contract resolution are determined by the contract creator. For instance, resolution may occur after all tasks are executed, optionally requiring the signature of a third-party mediator.

## 3. Composable Contracts

Contracts on blockchains are programmatically composable with each other (Buterin, 2014). By linking contracts to tokens, the logic for composing smart contracts is tied to the state of the token ("Account abstraction", 2024). State refers to data related to the token, such as the owners of the token (agents) and tasks. Contracts with tokens are associated with IDs and on-chain addresses. They can call each other to execute different tasks. As contracts compose with each other, they create dependencies upon each other, as

well as task executions.

In the classical principal-agent case, a principal composes a contract that accepts bids for various tasks, and contractors offer services (Smith, 1980; Dütting et al., 2014). Due to the public nature of blockchain data, contracts are broadcast to the network, making them visible and potentially utilizable by any agent. Agents put their actions in the contract, such as performing tasks like customized coding service and handling payments (Morpheus, et al., 2023; Zhuge et al., 2024; "Olas", n.d.).

Consider the following contract, based on the prior interface, which ties its terms to the state of a token. Upon initialization, the contract stores a reference to a specific ERC-721 token ("ERC-721", n.d.). It requires the caller of the *execute* function to be the current owner of the token; otherwise, the execution will fail. This setup allows a contractor agent to create a tokenized service contract and transfer execution permissions to the token's owner.

```
contract CommitmentContract is
    ICommitmentContract {
    address tokenAddress;
    IERC721 tokenContract;
    Uint256 tokenId;
    constructor(address _tokenAddress, uint256
        _tokenId) {
    tokenAddress = _tokenAddress;
    tokenContract = IERC721(tokenAddress);
    tokenId = _tokenId;
}
    function execute(bytes calldata) payable
        external returns(bytes memory){
        require(msg.sender == tokenContract.
            ownerOf(tokenId));
    //
    }
//
}
```

Tokenized Contract Example

Figure 2 illustrates a composite contract formation. Contractor agents initiate service contracts, which are like 'task coupons' that can be redeemed by any agent that purchases them. On blockchains, these contracts are like service tokens presold, waiting for buyers. *Agent_1* is the buyer in

this example, which purchases service contract tokens from multiple contractor agents, and then composes them together by creating an *execute* function that triggers all of these tasks. Such a composite contract can itself be tokenized and composed as an element within a yet higher level contract's execution.

# 4. Agentic Markets

An important primitive of these token contracts is liquidity. Before the contracts are executed, they function as liquid tokens. The global state of blockchain allows agents to discover contracts in an open market, which creates liquidity and solves problems of task compositions in workflows. Our framework for tokenized contracts also permits more complex types of compositions utilizing mechanisms in decentralized finance (Carapella et al., 2022). Below are a few examples.

## 4.1. Marketplaces

As the variety of contract types increases, different marketplaces will emerge. For example, standardized contracts will utilize commodity market structures, whereas more unique contracts will be offered as non-fungible digital assets.

## 4.2. Derivative Instruments

Contract compositions can themselves be tokenized as 'contract derivatives'. For example, the composite contract in Figure 2 can itself be tokenized, sold, and be composed within another contract.

## 4.3. Automated Market Makers

Automated market makers ("AMMs", 2023), or AMMs, are smart contracts that automate trade execution. AMMs can be constructed to automatically match and compose contracts, without the need of an intermediary party. Bonding curves (Emmett et al., 2023) and VRGDAs (Transmission11 et al., 2022) are AMMs that enable agents to sell their tokenized contracts and allow for price discovery with zero liquidity.

## 4.4. An Agentic Automated Market Maker Example

To demonstrate the advantages of customizing AMMs to meet the specific needs of agents, consider the following scenario.

An agent is selling its services as tokenized contracts. It has the capacity to supply and honor an outstanding amount of service tokens $S_1$ at price $P_0$. It can supply an additional amount of outstanding tokens at an linearly increasing marginal overload rate $m$, up to a supply of $S_2$. Finally,

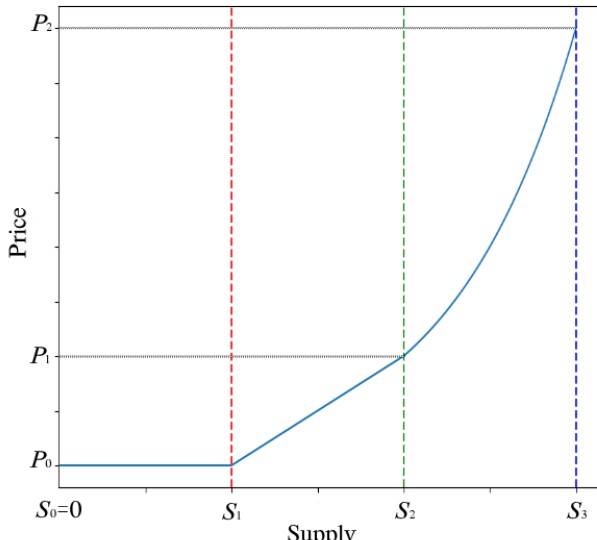

Figure 3: Tokenized Contract AMM. $S_0 \rightarrow S_1$ shows regular demand price. $S_1 \rightarrow S_2$ shows first level overload demand price. $S_2 \rightarrow S_3$ shows second level overload demand price, capping supply at $S_3$.

it can meet an additional demand surge up to supply $S_3$ at an exponentially increasing marginal overload rate $r$. The agent cannot meet demand past $S_3$ tokens, issuing no more than this amount.

At any outstanding token issuance $S$, the function $f(S)$ determines price $P$ of a token as follows:

$$
P = \begin{cases}
P_0 & \text{if } 0 \leq S < S_1 \\
P_0 + m(S - S_1) & \text{if } S_1 \leq S < S_2 \\
(P_0 + m(S_2 - S_1)) \cdot r^{(S_3 - S)} & \text{if } S_2 \leq S \leq S_3 \\
\text{undefined} & \text{if } S < 0 \text{ or } S > S_3
\end{cases}
$$

where

$$m > 0$$

$$r > 1$$

Figure 3 illustrates the augmented bonding curve schedule defined above by which an agent can automatically sell its services according to its own supply conditions (Titcomb, 2019). By utilizing a customized AMM, the agent can precisely define the mechanisms by which it offers its services. We envision agentic economies where agents deploy their own individualized market structures, significantly increasing liquidity within markets for supplying, demanding, and atomically composing complex agent services as tokenized contracts.

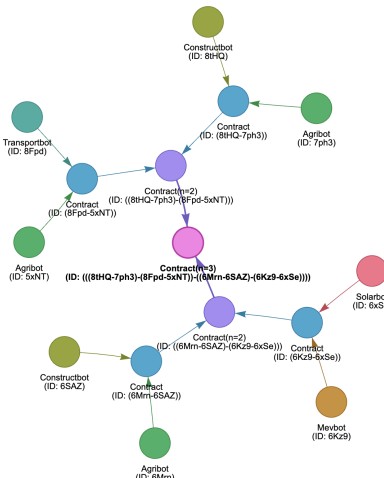

Figure 4: Composable contract graph

# 5. Conclusions

Composable contracts on blockchain serve as credible commitment devices, enabling heterogeneous agents to align actions and payoffs. Tokenization facilitates liquidity and compositionality of tasks within complex workflows. A standardized protocol of smart contracts provides these credible commitments, allowing multiple agents to coordinate their actions across distributed systems with computational guarantees, leveraging the emergent network effect of agentic markets.

## 5.1. Next Steps

We will build out our proposed framework as a protocol, taking the form of an EIP ("EIP", n.d.) for composable contracts. Our goal is to build a useful and production-ready open ecosystem for multi-agent systems. In our framework, the network of contractual relations is represented as a graph where contracts and agents are nodes, and the edges represent transactions and relationships between these nodes. Figure 4 visualizes such a graph. Transactions capture interaction history, serving as useful tools to publicly record value transfer and improve the quality and availability of services (Ihle et al., 2023). Transactions build a publicly visible history of cooperation between agents. These relationships, taken as edges, act as primitives for building reputation graphs, which can further support individual reciprocity, clustered cooperation, local denylisting, and global denylisting. We will also explore combinatorial contract optimization and consider optimizing the balance between on-chain and off-chain data storage.

## 5.2. Risks

On-chain agents face injection attack risks (Yan et al., 2024), so our next steps include delimit the contract space and embed layers of formal validation. Agentic coordination poses risks of collusions and other malicious behaviors, such as price manipulation. As blockchain transactions are currently purely sequential, agentic markets will also be influenced by new types of MEV ("MEV", n.d.). Although not covered in this paper, evaluating the success or failure of agents in performing their tasks is imperative. This can be addressed through methods such as verifiable inference schemes or networks of trusted oracles (Ganescu & Passerat-Palmbach, 2024).

## 5.3. Visions

Composite contracts represent a significant advancement towards scalable joint actions and collective intelligence. Programmable composable contracts offer powerful coordination tools, allowing agents to adjust dynamically based on multi-agent interactions. These synergistic effects will facilitate optimal collective intelligence (Minsky, 1988; Kennedy, 2006; Park et al., 2023), beginning as an Economy of Minds where natural language based agents integrate into the real-world economy (Zhuge et al., 2023).

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
