# OpenReview forum: "Composable Contracts for Multi-Agent Coordination"
_ICML.cc/2024/Workshop/Agentic_Markets — Agentic Markets @ ICML'24 Poster_

### Official Review · Reviewer_UJGQ · 2024-06-14
**Could not see any contribution whatsoever to the literature**

**Rating:** 5
**Confidence:** 3

**Review:**

The paper presents a framework for using ``blockchain-based, programmable contracts'' to ``improve coordination in multi-agent systems by reducing information asymmetry and ensuring reliable commitments''.

More simply, the main idea is that on one side, we have AI Agents that we'd like to endow with more autonomy, and on the other side, we have smart contracts on blockchain systems that enable transparency and pre-commitment of actions. The article is saying "wouldn't it be nice if we could link these two things, so that agents interact with each other via these open-source smart contracts that are verifiable".

The answer is sure, why not, but there doesn't seem to be anything deeper beyond this trivial idea. The article reads more like a high-level overview of what smart contracts can do on blockchain systems in general, and the AI aspect is largely missing from the discussion. I couldn't see any contribution to the literature unfortunately. In particular, the authors haven't had time to develop a demo system to showcase how it works, and why it's a contribution on top of what we already know.

The entire thing seems overly obvious / trivial to me. I wouldn't classify this as a research paper. It's more of a commentary on what existing systems are capable of.

---

### Official Review · Reviewer_NbGk · 2024-06-15
**Interesting proposal for blockchain-based multi-agent coordination but lacks novelty, implementation, or empirical support**

**Rating:** 5
**Confidence:** 2

**Review:**

**Summary:**
The paper explores using blockchain technology to enhance coordination in multi-agent systems. Through a framework of programmable, composable smart contracts, it aims to address issues of information asymmetry and ensure credible commitments among heterogeneous agents.

The paper lists some compelling reasons why the blockchain can be a good way to enforce collaboration between LLM agents, but beyond that, it reads to me more as an idea than a framework.

**Claims:**
- Blockchain technology can reduce informational friction among agents.
- Smart contracts provide modular, composable solutions for task allocation and payoff negotiation.
- Tokenized contracts and automated market makers can enhance liquidity and facilitate complex workflows in agentic markets.

**Strengths:**
- Concise and informative introduction highlighting the problems caused by information asymmetry and the need for credible commitments among heterogeneous agents.
- Clear articulation of the potential benefits of blockchain technology in multi-agent coordination.

**Weaknesses:**
- The paper largely reiterates known advantages of blockchain-based contracts, seemingly without any new mechanisms or empirical evidence.
- Lacks implementation strategies or realistic examples of the proposed framework.
- Vague on the specifics of what the proposed framework is and how it addresses the coordination challenges in multi-agent systems.

---

### Official Review · Reviewer_qriq · 2024-06-18
**2. #17 Composable Contracts for Multi-Agent Coordination**

**Rating:** 8
**Confidence:** 3

**Review:**

The paper starts with the non-intuitive proposition that when heterogeneous agents interact, there is necessarily information asymmetry. I understand that this may be a foregone conclusion in the field but it is not obvious to a layperson why this may be the case and the authors must do a better job of conveying it rather than redirecting the reader to past work. The paper then proposes a blockchain based solution with smart contracts focusing on the value it adds to shared information between agents accomplishing different intermediate steps in a sequence of activities in order to achieve a certain end goal.

While the paper highlights relevant aspects of why a blockchain based data storage and sharing mechanism makes sense to maintain information symmetry between heterogeneous agents, it really provides no explanation as to how a heterogeneous multiagent marketplace might contribute to information asymmetry in the first place. The solutions they provide seem to be more about pitching the blockchain's features as applied to agent-based information sharing given the lack of specific examples and lack of focus on specific facets of multiagent systems. In favor of the authors, I acknowledge that the paper is clear, the smart contract based mechanism is practical and feasible to apply to sequences of interactions in multiagent systems but it would be really helpful to motivate this with an example of a distinct problem that this is solving (e.g. a customer travel booking agent interacting with a number of websites for hotels and flights with their individual agents) instead of framing it as a general set of problems well-known enough to not even warrant a discussion in the paper. This would elevate the value of the theory proposed in the paper.
Line 031 has a typo (should be agents', or an agent's).

I recommend acceptance as a poster (4/5) with medium confidence.